# A new ferrocene derivative blocks K-Ras localization and function by oxidative modification at His95

Kristen M Rehl[1], Jayaraman Selvakumar[2], Rhonda L Pitsch[3], Don Hoang[2], Kuppuswamy Arumugam[2],
Sean W Harshman[3], Alemayehu A Gorfe[4], Kwang-Jin Cho[1]

Ras proteins are membrane-bound GTPases that regulate essential cellular processes at the plasma membrane (PM). Constitutively active mutations of K-Ras, one of the three Ras isoforms in mammalian cells, are frequently found in human cancers. Ferrocene derivatives, which elevate cellular reactive oxygen species (ROS), have shown to block the growth of non-small cell lung cancers harboring oncogenic mutant K-Ras. Here, we tested a novel ferrocene derivative on the growth of pancreatic ductal adenocarcinoma and non-small cell lung cancer. Our compound, which elevated cellular ROS levels, inhibited the growth of K-Ras-driven cancers, and abrogated the PM binding and signaling of K-Ras in an isoform-specific manner. These effects were reversed upon antioxidant supplementation, suggesting a ROS-mediated mechanism. We further identified that K-Ras His95 residue plays an important role in this process, and it is putatively oxidized by cellular ROS. Together, our study demonstrates that the redox system directly regulates K-Ras/PM binding and signaling via oxidative modification at the His95, and proposes a role of oncogenic mutant K-Ras in the recently described antioxidant-induced growth and metastasis of K-Ras-driven cancers.

## Introduction

Ras proteins are a family of small GTPases primarily localized to the plasma membrane (PM) and function as molecular switches, cycling between a GDP-bound inactive state and a GTP-bound active state (McCormick, 2016; Chen et al, 2021). In response to activation by upstream receptor kinases, guanine exchange factors bind to Ras to facilitate the release of GDP and binding of GTP, which confers Ras the active conformation for binding its downstream effectors (McCormick, 2016; Chen et al, 2021). Ras proteins are involved in a variety of cellular pathways that regulate cell differentiation, proliferation, and survival (Jancik et al, 2010; Chen et al, 2021). GTPase-activating proteins (GAPs) promote Ras GTPase activity, enhancing Ras-induced GTP hydrolysis to GDP (McCormick, 2016; Chen et al, 2021). There are three Ras isoforms that are ubiquitously expressed in mammalian cells: K-, H-, and N-Ras. K-Ras undergoes alternative splicing on the fourth exon, yielding K-Ras4A and K-Ras4B, where K-Ras4B, hereafter K-Ras, is the predominant K-Ras protein expressed in mammalian cells (Cox et al, 2014). Constitutively active mutations of Ras are found in ~19% of all human cancers, with ~75% of those being K-Ras (Prior et al, 2020). Oncogenic mutant K-Ras are found in ~88% pancreatic, ~50% colorectal, and ~30% of lung cancers (Prior et al, 2020), and a group of newly developed K-Ras G12C direct inhibitors have shown promise in clinical trials. Sotorasib and adagrasib are FDA-approved small molecules that directly bind to the GDP-bound K-Ras G12C mutant and form a covalent bond to the mutant Cys, which locks K-Ras in the inactive conformation, thereby blocking its signaling (Ostrem et al, 2013; Canon et al, 2019; Hallin et al, 2020). Although these inhibitors exhibit pronounced anti-cancer effects in K-Ras G12C tumor mice models and in clinical trials (Canon et al, 2019; Hallin et al, 2020), the K-Ras G12C mutation is found only in ~3% pancreatic, ~4% colorectal, and ~13% of lung cancers that harbor any oncogenic mutations in K-Ras, making it effective against only a small subset of the K-Ras-driven cancers (Cox et al, 2014; Prior et al, 2020).

One of the strategies for blocking pan-oncogenic mutant K-Ras activity is to dissociate it from the PM. Experimental data show that K-Ras, regardless of mutation status, activates its downstream effectors primarily at the inner PM leaflet, and that dissociating K-Ras from the PM blocks K-Ras signal output and the growth of K-Ras-driven cancers (Cho et al, 2012b, 2016a, 2016b; van der Hoeven et al, 2013; Kattan et al, 2019, 2021; Miller et al, 2019; Tan et al, 2019; Garrido et al, 2020; Kovar et al, 2020). Ras proteins have two targeting signals for stable interaction with the PM. The first signal is the C-terminal CAAX motif that undergoes a series of posttranslational modifications to generate a farnesyl–cysteine–methyl–ester, allowing Ras binding to cellular membranes (Apolloni et al, 2000; Prior & Hancock, 2012). For H- and N-Ras, the second targeting

[1]Department of Biochemistry and Molecular Biology, Boonshoft School of Medicine, Wright State University, Dayton, OH, USA  [2]Department of Chemistry, College of Science and Mathematics, Wright State University, Dayton, OH, USA  [3]Air Force Research Laboratory, Wright-Patterson AFB, OH, USA  [4]Department of Integrative Biology and Pharmacology, McGovern Medical School, University of Texas Health Science Center, Houston, TX, USA

Correspondence: Kwang-jin.cho@wright.edu

signal is palmitoyl moieties adjacent to the farnesylated Cys, conferring their stable PM binding. The second targeting signal of K-Ras is its C-terminal polybasic domain, a stretch of six lysine residues near the farnesyl moiety, which forms favorable electrostatic interactions with phosphatidylserine (PtdSer), an anionic phospholipid enriched in the PM inner leaflet (Hancock et al, 1990; Yeung et al, 2008; Zhou et al, 2017). Recent studies have identified three molecular mechanisms that regulate K-Ras transport to, and interaction with the PM (i) disrupting K-Ras binding to its chaperone protein, phosphodiesterase 6 delta (PDE6$\delta$), (ii) phosphorylating K-Ras at Ser181, and (iii) reducing PM PtdSer content (Gorfe & Cho, 2019; Henkels et al, 2021). Under basal conditions, H-, N-, and K-Ras proteins are farnesylated, and PDE6$\delta$ continuously sequesters Ras proteins from endomembranes via interacting with the prenyl moiety and returns them to the PM. Disrupting this interaction redistributes all three Ras isoforms to endomembranes (Chandra et al, 2012; Zimmermann et al, 2013; Schmick et al, 2014). Also, protein kinase C and G directly phosphorylate K-Ras, but not other Ras isoforms, at Ser181 via independent mechanisms. This perturbs K-Ras PM binding and mislocalizes K-Ras from the PM to endomembranes (Bivona et al, 2006; Cho et al, 2016a; Kovar et al, 2020). The other regulator for K-Ras PM localization is PtdSer enrichment in the inner PM leaflet. Depletion of PM PtdSer by perturbing mechanisms that maintain PM PtdSer abundance dissociates K-Ras from the PM and blocks K-Ras signal output and the growth of human cancers harboring oncogenic mutant K-Ras in vitro and in vivo (Cho et al, 2012b, 2016b; van der Hoeven et al, 2018; Kattan et al, 2021).

Ferrocene derivatives encompass a diverse range of compounds, but they all contain the ferrocene molecule, an iron atom (Fe) sandwiched between two cyclopentadienyl ligands (Kealy & Pauson, 1951; Wilkinson et al, 1952; Gasser et al, 2011). The ferrocene molecule is insoluble and inactive, requiring additional functional groups to enter the cell (Larik et al, 2016; Peter & Aderibigbe, 2019). Once inside, it undergoes a one-electron oxidation, $Fe^{2+}$ to $Fe^{3+}$, to form the active ferrocenium cation (Gasser et al, 2011; Larik et al, 2016). The general mechanism of ferrocene derivatives involves the elevation of cellular reactive oxygen species (ROS) to induce cellular damage and apoptosis (Melendez, 2012; Mojzisova et al, 2014). For example, a ferrocene conjugated to a gold(I)N-heterocyclic carbene complexes elevates cellular ROS by irreversibly binding to and inhibiting thioredoxin reductase, an essential component of the thioredoxin antioxidant system (Arambula et al, 2016). Human cancers harboring oncogenic mutant K-Ras are susceptible to changes in cellular ROS because of their metabolic reprogramming, and ROS-elevating agents are effective in their growth inhibition (Arambula et al, 2016; McCall et al, 2017; Foo & Pervaiz, 2019). Also, ferrocene derivatives exhibit anti-cancer effects in lung cancer cells harboring oncogenic mutant K-Ras (Domarle et al, 1998; Perez et al, 2015; Arambula et al, 2016; Peter & Aderibigbe, 2019). Thus, ferrocene derivatives may be useful for targeting K-Ras-driven cancers.

In this study, we developed a novel ferrocene complex, $C_{16}H_{20}FeClNO$, which inhibits the growth of K-Ras-driven human pancreatic ductal adenocarcinoma (PDAC) and non-small cell lung cancer (NSCLC) cells. We also discovered that elevation of cellular ROS disrupts the PM binding and signaling of K-Ras, but not other

Ras isoforms, through oxidative modification of K-Ras His95 residue. Moreover, our findings propose an additional role of oncogenic mutant K-Ras in the recently reported antioxidant-induced growth and metastasis of K-Ras-driven NSCLC (Sayin et al, 2014; Lignitto et al, 2019; Wiel et al, 2019).

# Results

## Synthesis of $C_{16}H_{20}FeClNO$

Our group recently reported a set of chalcones bearing $\alpha$, $\beta$-unsaturated carbonyl motifs that blocks the growth of PDAC and NSCLC cell lines harboring oncogenic mutant K-Ras by phosphorylating K-Ras at Ser181 (Kovar et al, 2020). Also, ferrocene derivatives have shown to inhibit the growth of NSCLC harboring oncogenic mutant K-Ras via mechanisms that have not been elucidated (Arambula et al, 2016; Larik et al, 2016; Wang et al, 2020). Thus, we envisaged that an ideal drug to target K-Ras-driven cancers could be designed by combining the $\alpha$, $\beta$-unsaturated carbonyl and ferrocene motifs. To test this, we devised an elegant strategy to fuse these two motifs in a single-pot reaction. Also, we included an ammonium group to increase the solubility in an aqueous solution. Compound **1** was prepared by treating acetylferrocene with a premixed solution of bis(dimethylamino)methane, phosphoric, and acetic anhydride (Fig 1A). After workup, compound **1** was obtained as a crude product, which was purified using column chromatography to yield the pure product as an orange–yellow oil in 71% yield. The obtained product was characterized using [1]H and [13]C NMR spectroscopy to confirm the identity of compound **1** (Fig S1A and B). The resulting oil was further treated with HCl dissolved in diethyl ether to yield pure compound **2** in 95% yield (Fig 1A). The obtained product was subjected to [1]H and [13]C NMR spectroscopy, and the identity was confirmed using high-resolution mass spectroscopy (MS) and elemental analysis (Fig S1C and D). The obtained compound **2** ($C_{16}H_{20}FeClNO$) was further subjected to biological studies.

## Compound 2 inhibits the growth of K-Ras-dependent human cancers

A wide range of human cancers harboring oncogenic mutant K-Ras reprogram their signaling network so that their survival and growth depend on the oncogenic K-Ras signaling, a phenomenon called K-Ras addiction or dependence (Weinstein & Joe, 2008; Singh et al, 2009, 2012; Hayes et al, 2016). To examine if our compound **2** inhibits the growth of K-Ras-dependent human cancers, a panel of human PDAC and NSCLC cell lines, in which their growth are K-Ras-dependent or -independent, were grown with compound **2**, and cell proliferation was measured. Our data show that the growth of K-Ras-dependent cancer cell lines was more sensitive than K-Ras-independent control cell lines (Fig 1B–E). We further examined Ras signaling by measuring phosphorylated ERK (ppERK) and Akt (pAkt), the two most well-studied Ras downstream effectors, after treating selected PDAC and NSCLC cell lines with high sensitivity for compound **2**. Our immunoblot data show that although there is a dose-

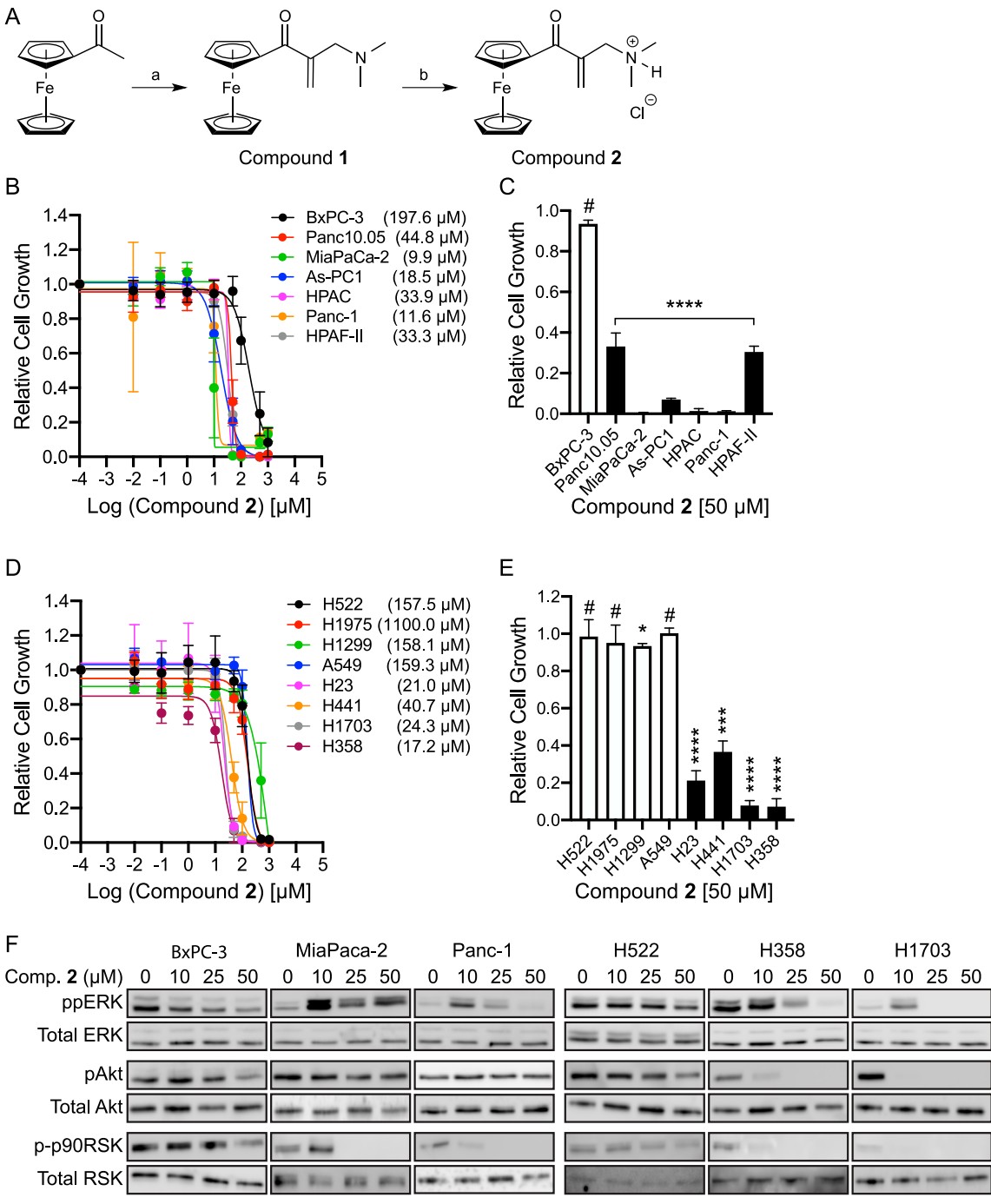

**Figure 1. Compound 2 blocks the growth of K-Ras-dependent human PDAC and NSCLC cells.**
**(A)** Systematic synthesis of ferrocene derivatives. **a** = $CH_2(N(CH_3)_2)_2$, $H_3PO_4$, $CH_3COOH$, 110°C, 4 h; **b** = HCl in $O(CH_2CH_3)_2$, $CH_2Cl_2$, 0–23°C. **(B, D)** A panel of PDAC and NSCLC cells were plated on 96-well plates and treated with various concentrations of compound **2** for 3 d. Complete growth medium with the compound was replaced every 24 h. Cell proliferation was analyzed using CyQUANT proliferation assay. The graphs show the mean cell number ± SEM from three independent experiments relative to that for control cells (DMSO-treated). Each value represents an $IC_{50}$ estimated from the dose–response plots generated by variable slopes (four parameters) in Prism software. **(C, E)** The graphs show the mean cell number ± SEM relative to that for the control cells (DMSO-treated cells) after 50 $\mu$M treatment from (B, D). Open and closed bars represent cell lines, in which their growths are K-Ras-independent and -dependent, respectively. Significant differences between control (DMSO-treated) and compound **2**-treated cells were assessed using $t$ tests (*$P < 0.05$, ***$P < 0.001$, ****$P < 0.0001$, # not significant). **(F)** Cell lysates from PDAC and NSCLC cell lines treated with various concentrations of compound **2** for 24 h were immunoblotted for ppERK, pAkt (S473) and p-p90RSK. Total ERK, Akt, and RSK blots were used as loading controls. Representative blots from three independent experiments are shown.

dependent reduction of ppERK in BxPC-3 and H522, control cell lines for PDAC and NSCLC, respectively, ppERK level was initially elevated at 10 $\mu$M and showed a trend of decrease at higher concentrations in K-Ras-dependent cell lines (Fig 1F). To further evaluate the ppERK activity, we measured the phosphorylation of p90 ribosomal S6 kinase (RSK), the cytosolic substrate of ERK1/2 (Hayes et al, 2016). Our data show that although p90RSK phosphorylation followed the trend of ppERK in control cell lines, it was decreased from 10 $\mu$M, the concentration that elevated ERK phosphorylation, in K-Ras-dependent cancer cell lines (Fig 1F). A similar observation has been reported, in which ERK inhibitor SCH772984 paradoxically elevates ERK phosphorylation, whereas it reduces p-p90RSK levels in PDAC. This study proposes that kinome reprogramming overcomes the suppression of MEK-induced ERK phosphorylation by the inhibitor, but not ppERK activity (Hayes et al, 2016). Together, our data suggest that compound **2** has a higher selectivity for the growth inhibition of K-Ras-dependent PDAC and NSCLC, and that the compound may disrupt the kinome reprogramming, resulting in elevated ERK phosphorylation, but blocks ppERK activity.

## Compound 2 translocates K-Ras from the plasma membrane to endomembranes

K-Ras conducts signal transduction by stimulating its direct downstream effectors primarily at the PM, and blocking the PM localization abrogates K-Ras signaling and the growth of K-Ras-dependent cancers (Cho et al, 2012b, 2016a, 2016b; van der Hoeven et al, 2018; Miller et al, 2019; Garrido et al, 2020; Kovar et al, 2020). To examine if compound **2** perturbs K-Ras/PM binding, we directly measured K-Ras/PM binding by quantitative electron microscopy (EM). Intact basal PM sheets from MDCK cells stably expressing GFP-tagged oncogenic mutant K-Ras (K-RasG12V) or -H-RasG12V and treated with the compound for 48 h were labeled with anti-GFP antibody conjugated to 4.5-nm gold particles and imaged by EM (Prior et al, 2001, 2003). Our data show that the total number of gold particles for GFP-K-RasG12V, but not -H-RasG12V, was significantly decreased after compound **2** treatment (Figs 2A and S2). Ras proteins in the PM are spatially organized into nanoscale domains, called nanoclusters, which are essential for high-fidelity Ras signal transduction (Prior et al, 2003; Tian et al, 2007; Cho et al, 2012a). Thus, we examined the spatial organization of Ras proteins remaining at the PM after compound **2** treatment. Our data show a significant reduction in the nanoclustering of K-RasG12V, but not H-RasG12V, after compound **2** treatment (Fig 2B), suggesting that compound **2** disrupts the PM binding and nanoclustering of K-Ras, but not H-Ras.

To further investigate the cellular localization of K-RasG12V after compound **2** treatment, MDCK cells stably co-expressing mCherry-tagged CAAX, a generic endomembrane marker (Choy et al, 1999; Cho et al, 2012b), and GFP-K-RasG12V were treated with various concentrations of compound **2** for 48 h and imaged by confocal microscopy. To quantitate the extent of K-Ras translocation from the PM to endomembranes, an IC$_{50}$ value for Ras/PM dissociation was derived from Manders' coefficient, which calculates the fraction of mCherry-CAAX co-localized with GFP-K-RasG12V (Manders et al, 1993; Cho et al, 2012b). Our data show that compound **2** translocated K-RasG12V to endomembranes with an IC$_{50}$ of 0.39 $\mu$M (Fig 2C and E). Our confocal microscopy further shows that the

compound did not disrupt the PM localization of GFP-H-RasG12V, -N-RasG12V, and -K-Ras4A G12V (Fig 2D and E). Subcellular fractionation assay further reveals that the membrane-associating K-RasG12V is reduced, whereas the cytosolic K-RasG12V is elevated after compound **2** treatment. No changes were observed with H-RasG12V (Fig 2F and G). We repeated the fractionation assay in wild-type (WT) Panc-1 cell line and measured membrane-bound and cytosolic endogenous Ras isoforms. Our data show that the membrane-bound endogenous K-Ras4B, but not other Ras isoforms, is reduced (Fig 2H).

To examine to which cellular organelles the mislocalized K-Ras is redistributed, we co-stained cells expressing GFP-K-RasG12V with organelle markers after compound **2** treatment. Our data show that K-RasG12V co-localized with Rab5A-, Rab7A-, LAMP1-, GALNT1-, and ER-tracker–positive structures after the treatment (Fig S3A–D and F), suggesting that compound **2** translocates K-Ras to the early endosome, late endosome, lysosome, the Golgi complex, and ER. K-RasG12V did not co-localize with a mitochondrial protein, PDHA1, indicating K-RasG12V did not translocate to the mitochondria (Fig S3E). Together, our data suggest that compound **2** disrupts K-Ras/PM binding and nanoclustering of the remaining K-Ras at the PM, but not other Ras isoforms, and that the mislocalized K-Ras translocates to the cytosol and various cellular organelles except the mitochondria.

## Compound 2 blocks the K-Ras/MAPK signaling

To investigate if the isoform-specific inhibitory effect on the Ras/PM localization translates to Ras signaling, MDCK cells expressing GFP-tagged different oncogenic mutant Ras isoforms were treated with compound **2** and probed for ppERK and pAkt (S473). Our data show that compound **2** significantly decreased the level of ppERK, but not pAkt, in a dose-dependent manner only in K-RasG12V–expressing cells (Fig 3A and B). A previous study reported that K-Ras is a more potent activator of the Raf-1/MAPK signaling than H-Ras, whereas H-Ras is a more potent activator of the PI3K/Akt signaling (Yan et al, 1998). Thus, it is plausible that compound **2** blocks the PM binding and nanoclustering of K-Ras, but not other Ras isoforms, resulting in significant inhibition of the Raf-1/MAPK signaling with minimal effects on the PI3K/Akt signaling. Also, we showed that compound **2** differentially regulates ERK phosphorylation in human cancer cell lines (Fig 1F), another evidence of a more complex signaling network in human cancer cell lines compared with a model cell line.

We also measured total K-Ras protein expression because K-Ras/PM dissociation by various mechanisms differentially regulate K-Ras protein expression (Cho et al, 2012b; van der Hoeven et al, 2013; Garrido et al, 2020). Our data show that the compound did not change the protein expression of Ras isoforms (Fig 3A and B). Moreover, compound **2** did not induce cleavage of caspase-3 and PARP-1, proteins involved in apoptosis (Fig S4), which suggest that the compound does not induce apoptosis at the concentration that blocks K-Ras/PM binding and K-Ras/MAPK signaling. Together with Fig 2, our data suggest that compound **2** inhibits the PM binding, nanoclustering, and MAPK signaling of K-Ras, but not other Ras isoforms.

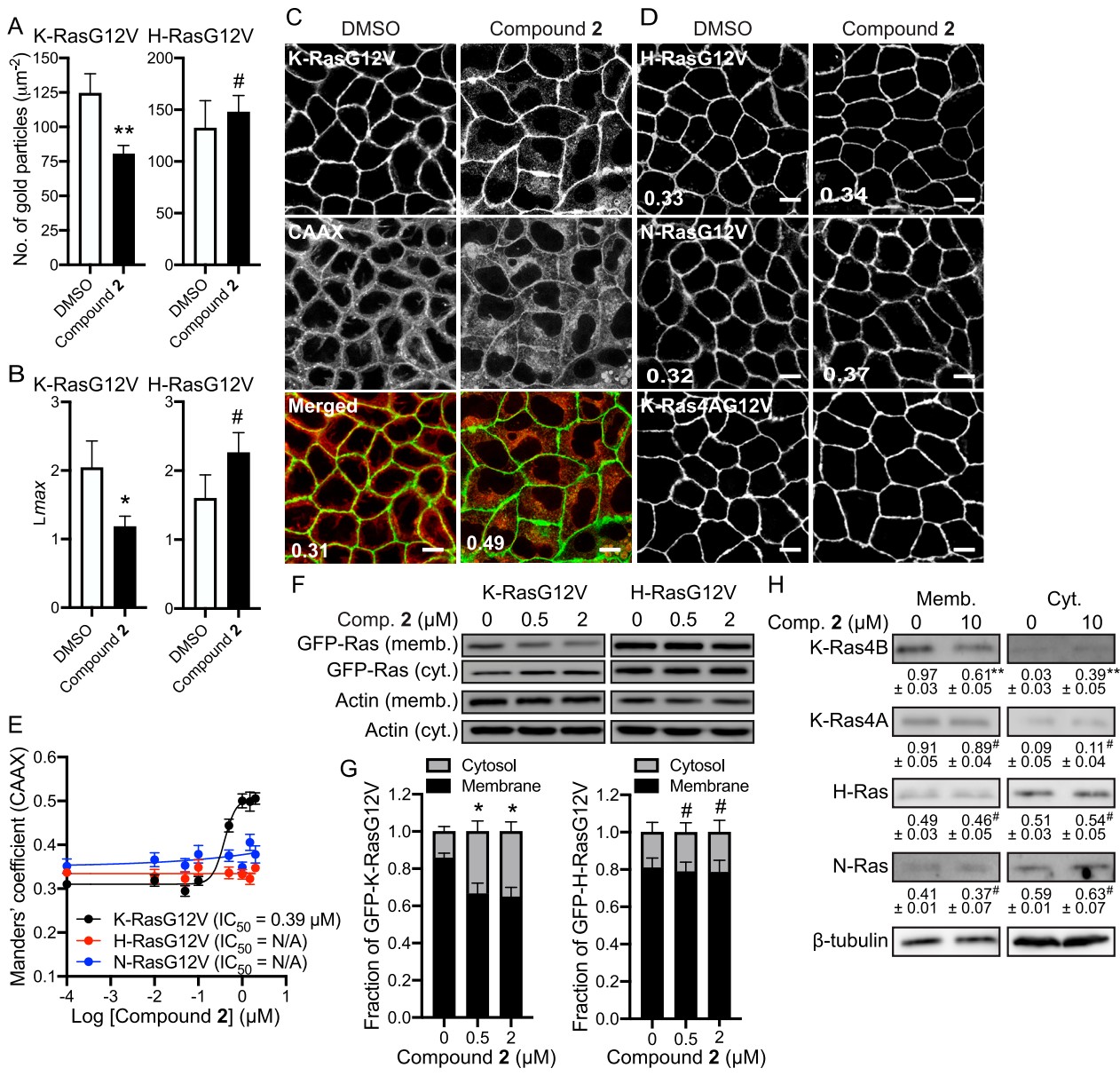

**Figure 2. Compound 2 translocates K-Ras from the PM to the cytosol and endomembranes.**
**(A)** Basal PM sheets prepared from MDCK cells expressing GFP-K-RasG12V or -H-RasG12V and treated with 2 μM compound **2** for 48 h were labeled with anti-GFP–conjugated gold and visualized by EM. Representative EM images are shown in (Fig S1). The graphs show a mean number of gold particles ± SEM (n ≥ 15). Significant differences between control (DMSO-treated) and compound **2**-treated cells were assessed by t tests. **(B)** Spatial mapping of the same gold-labeled PM sheets was performed. The peak values, $L_{max}$, of the respective weighted mean K-function $L(r)-r$ curves are shown as bar graphs (n ≥ 15). Significant differences between control (DMSO-treated) and compound **2**-treated cells were evaluated with bootstrap tests (*$P < 0.05$, **$P < 0.01$, #, not significant). **(C, D)** MDCK cells stably co-expressing mCherry-CAAX and (C) GFP-K-RasG12V, (D) -H-RasG12V or -N-RasG12V, or expressing GFP-K-Ras4A G12V only were treated with various concentrations of compound **2** for 48 h. Cells were fixed with 4% PFA and imaged by confocal microscopy. Representative images of 2 μM compound **2**-treated cells are shown. Inserted values represent an estimated mean fraction of mCherry–CAAX co-localizing with GFP-K-RasG12V, -H-RasG12V or -N-RasG12V calculated by Manders' coefficient from three independent experiments. Scale bar: 10 μm. **(E)** $IC_{50}$s were estimated from the dose–response plots. **(F)** MDCK cells stably expressing GFP-K-RasG12V or -H-RasG12V were treated with compound **2** for 48 h. Cell lysates were fractionated into membrane (memb.) and cytosol (cyt.) fractions. 5 and 20 μg protein for membrane and cytosolic fractions, respectively, were immunoblotted to measure GFP-RasV12 level using an anti-GFP antibody. Representative blots from three independent experiments are shown. Actin blots were used as loading controls. **(G)** The graphs show the mean membrane-bound fraction ± SEM of RasG12V, calculated as membrane/(membrane + cytosol) from three independent experiments. **(H)** Panc-1 cells were treated with 10 μM compound **2** for 24 h. Cell lysates were fractionated into membrane and cytosol fractions. Endogenous Ras isoforms were immunoblotted using Ras isoform-specific antibodies. Different protein amounts were used to measure membrane-bound and cytosolic K-Ras4B (5 and 20 μg, respectively), K-Ras4A (5 and 20 μg), H-Ras (10 and 20 μg), and N-Ras (10 and 20 μg). Representative blots from three independent experiments are shown. β-tubulin blots were used as loading controls. **(G, H)** Significant differences of membrane fraction between control (DMSO-treated) and compound **2**-treated cells were assessed by one-way ANOVA tests for (G) and t test for (H) (*$P < 0.05$, **$P < 0.01$, #, not significant).

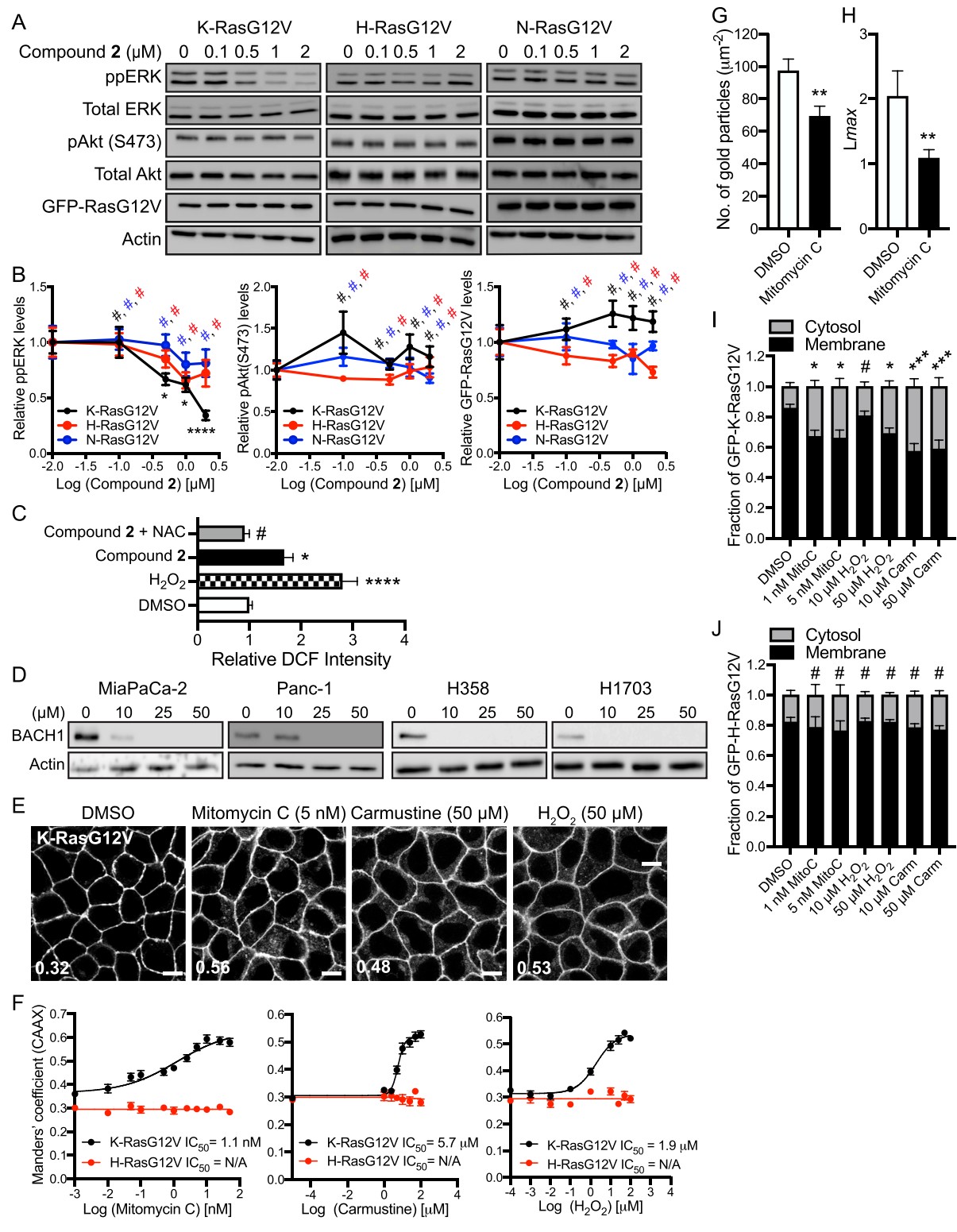

**Figure 3. Compound 2 blocks the K-Ras/MAPK signaling, and ROS-elevating agents dissociate K-Ras from the PM.**
**(A)** MDCK cells stably expressing GFP-K-RasG12V, -H-RasG12V or -N-RasG12V were treated with compound **2** for 48 h. Cell lysates were immunoblotted for ppERK, pAkt (S473), and GFP-RasG12V. Actin, total ERK, and Akt blots are shown as loading controls. Representative blots from three independent experiments are shown. **(B)** The graphs show the relative mean of ppERK, pAkt (S473), and GFP-RasG12V ± SEM from three independent experiments. **(C)** WT MDCK cells were seeded onto 96-well plates and grown in the presence 25 $\mu$M DCFH$_2$-DA with H$_2$O$_2$ (100 $\mu$M), compound **2** (2 $\mu$M), and/or NAC (500 $\mu$M) for 48 h. DCF, a readout for cellular ROS was measured by a fluorescent plate reader using emission = 495 nm and excitation = 527 nm. **(D)** K-Ras-dependent PDAC and NSCLC cell lines were treated with various concentrations of compound **2** for 24 h and cell lysates were immunoblotted for BACH1. Representative blots from three independent experiments are shown. Actin blots were used as

## Compound 2 disrupts K-Ras/PM binding and signaling via elevating cellular ROS

Because ferrocene derivatives elevate cellular ROS levels (Acevedo-Morantes et al, 2012; Arambula et al, 2016), we examined if our compound elevates cellular ROS. Briefly, WT MDCK cells were treated with compound **2** in the presence of DCFH$_2$-DA (2′,7′-dichlorodihydrofluorescein diacetate), a chemically reduced form of fluorescein. Once it enters a cell, esterases convert it to DCFH, which is further metabolized to highly fluorescent DCF upon exposure to cellular ROS, a readout for cellular ROS levels (Acevedo-Morantes et al, 2012). Our data show that compound **2** increased DCF production, whereas co-treatment with N-acetylcysteine (NAC), a general antioxidant (Spagnuolo et al, 2006) reversed it (Fig 3C), confirming that compound **2** elevates cellular ROS at the concentration that blocks K-Ras/PM binding and signaling. Previous studies have reported that under oxidative stress, the oxidation of heme-containing proteins releases free heme, which stimulates degradation of BACH1 (BTB and CNC homology 1), a pro-metastatic transcriptional factor (Zenke-Kawasaki et al, 2007; Lignitto et al, 2019). To examine if compound **2** induces cellular oxidative stress, we measured BACH1 protein level after treating K-Ras-dependent PDAC and NSCLC cell lines with compound **2**. Our data show that BACH1 was reduced from 10 $\mu$M, indicative of elevated cellular oxidative stress (Fig 3D). These data suggest that compound **2** elevates cellular ROS and oxidative stress.

To validate the role of cellular ROS in K-Ras/PM binding, we tested other chemical modulators that elevate cellular ROS by blocking the glutathione or thioredoxin antioxidant systems. MDCK cells co-expressing mCherry-CAAX and GFP-K-RasG12V or -H-RasG12V were treated with mitomycin C, a thioredoxin reductase inhibitor, carmustine, a glutathione reductase inhibitor or hydrogen peroxide ($H_2O_2$) (An et al, 2011; Paz et al, 2012; Yokoyama et al, 2017), and cellular localization of Ras proteins were imaged. Our data show that these compounds mislocalized K-RasG12V, but not H-RasG12V (Figs 3E and F and S5C). EM analysis further shows that mitomycin C blocks K-Ras/PM binding, nanoclustering, and K-Ras/MAPK signaling (Figs 3G and H and S5A and B). Moreover, subcellular fractionation assay reveals that mitomycin C, carmustine, and a high concentration of $H_2O_2$ significantly increased the cytosolic fraction of GFP-K-RasG12V, but not H-RasG12V (Fig 3I and J). Together, our data suggest that compound **2** elevates cellular ROS, which disrupts the PM binding and signaling of K-Ras, but not H-Ras.

## N-acetylcysteine reverts the effects of compound 2

To further validate that compound **2** disrupts K-Ras/PM binding and signaling by elevating cellular ROS, we repeated these experiments after NAC supplementation. MDCK cells expressing GFP-K-RasG12V were co-treated with NAC and compound **2**, mitomycin C or $H_2O_2$, and cellular localization of K-RasG12V was imaged. Our confocal and electron microscopy show that NAC supplementation restored K-Ras/PM binding and nanoclustering (Fig 4A–D). Our immunoblot data further show that co-treatment of compound **2** with NAC rescued the abrogated ppERK level in K-RasG12V-expressing cells (Fig 4E). Moreover, NAC supplementation restored the impaired growth of K-Ras-dependent PDAC and NSCLC cell lines (Fig 4F). Of note, although NAC alone elevates K-Ras/PM binding and ERK phosphorylation in K-RasG12V-expressing MDCK cells, it did not promote the growth of K-Ras-dependent cancer cells (Fig 4C, E, and F). Together, our data suggest that compound **2** blocks K-Ras/PM binding, nanoclustering, signaling, and the growth of K-Ras-dependent cancer cells via an ROS-mediated mechanism.

### ROS-induced K-Ras/PM dissociation is independent of PM PtdSer content and K-Ras Ser181 phosphorylation

PtdSer is an anionic phospholipid enriched in the inner leaflet of the PM and plays a critical role in the stable K-Ras/PM binding. K-Ras stably binds to the PM via the C-terminal farnesyl moiety in conjugation with the electrostatic interaction between the polybasic domain and the anionic head group of PtdSer (Yeung et al, 2008; Cho et al, 2012b; Zhou et al, 2017). Depletion of PM PtdSer content dissociates K-Ras from the PM and abrogates K-Ras signal output (Cho et al, 2012b, 2016b; Tan et al, 2018; van der Hoeven et al, 2018; Kattan et al, 2019, 2021). Another mechanism regulating the stable K-Ras/PM binding is K-Ras phosphorylation at Ser181 by protein kinase C or G. Stimulated protein kinase C and G phosphorylate K-Ras Ser181 residue, located adjacent to the polybasic domain, which perturbs the electrostatic interaction of K-Ras and the negatively charged cellular membranes, resulting in K-Ras/PM dissociation (Bivona et al, 2006; Cho et al, 2016a; Kovar et al, 2020). To examine if compound **2**-induced K-Ras/PM dissociation is via these mechanisms, we studied cellular distribution of GFP-LactC2 (the C2 domain of lactadherin), a well-studied PtdSer probe (Yeung et al, 2008) and -K-RasG12V S181A, where Ser181 is substituted to Ala to prevent it from being phosphorylated. Our confocal microscopy shows that compound **2** treatment had minimal effects on the PM localization of LactC2, whereas K-RasG12V S181A was redistributed

---

loading controls. **(E)** MDCK cells co-expressing mCherry-CAAX and GFP-K-RasG12V or -H-RasG12V were treated with various concentrations of ROS-elevating agents for 48 h, and imaged by confocal microscopy. Representative images of GFP-RasG12V are shown. Inserted values represent an estimated mean fraction of mCherry–CAAX co-localized with GFP-RasG12V calculated by Manders' coefficient from three independent experiments. Scale bar: 10 $\mu$m. **(F)** IC$_{50}$s were estimated from the dose–response plots. **(G)** Basal PM sheets prepared from MDCK cells expressing GFP-K-RasG12V and treated with 5 nM mitomycin C for 48 h were labeled with anti-GFP-conjugated gold and visualized by EM. The graph shows a mean number of gold particles ± SEM (n ≥ 15). **(H)** Spatial mapping of the same gold-labeled PM sheets was performed. The peak values, $L_{max}$, of the respective weighted mean K-function $L(r)—r$ curves are shown as bar graphs (n ≥ 15). **(I, J)** MDCK cells stably expressing GFP-K-RasG12V (I) or -H-RasG12V (J) were treated with indicated drugs for 48 h. Cell lysates were fractionated into membrane and cytosol fractions, and RasG12V levels were measured by immunoblotting using an anti-GFP antibody. The graphs show the mean membrane-bound fraction ± SEM of RasG12V, calculated as membrane/(membrane + cytosol) from three independent experiments. **(B, C, G, H, I, J)** Significant differences between control (DMSO-treated) and drug-treated cells were evaluated by one-way ANOVA tests for (B, C, I, J), $t$ tests for (G), and bootstrap tests for (H) (*$P < 0.05$, **$P < 0.01$, ***$P < 0.001$, ****$P < 0.0001$, #, not significant).

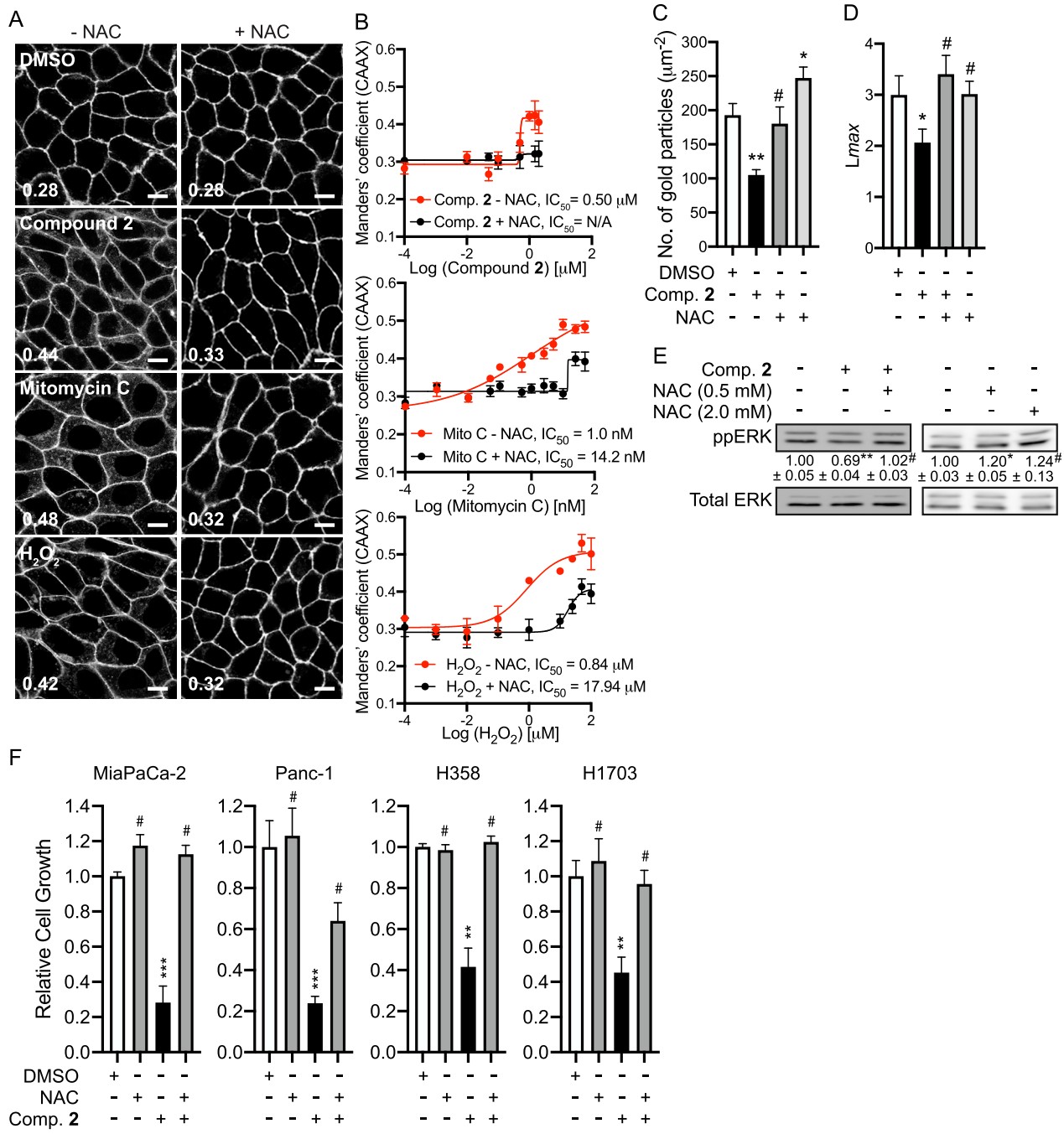

**Figure 4. NAC supplementation returns K-Ras to the PM.**

**(A)** MDCK cells co-expressing GFP-K-RasG12V and mCherry-CAAX were treated with 2 $\mu$M compound **2**, 5 nM mitomycin C, 50 $\mu$M H$_2$O$_2$ in the presence or absence of 500 $\mu$M NAC for 48 h, and imaged by confocal microscopy. Representative images of GFP-RasG12V are shown. Inserted values represent an estimated mean fraction of mCherry-CAAX co-localized with GFP-RasG12V calculated by Manders' coefficient from three independent experiments. Scale bar: 10 $\mu$m. **(B)** IC$_{50}$s were estimated from the dose–response plots. **(C)** Basal PM sheets prepared from MDCK cells expressing GFP-K-RasG12V and treated with 2 $\mu$M compound **2** alone, 500 $\mu$M NAC alone or together for 48 h were labeled with anti-GFP-conjugated gold and visualized by EM. The graph shows a mean number of gold particles ± SEM (n ≥ 15). **(D)** Spatial mapping of the same gold-labeled PM sheets was performed. The peak values, $L_{max}$, of the respective weighted mean K-function $L(r)-r$ curves are shown as bar graphs (n ≥ 15). **(E)** MDCK cells expressing GFP-K-RasG12V were treated with 2 $\mu$M compound **2** with or without 500 $\mu$M NAC for 48 h. Cell lysates were blotted for ppERK. Values indicate the mean ppERK ± SEM from three independent experiments. Representative blots are shown. Total ERK and actin blots are shown as loading controls. **(F)** K-Ras-dependent PDAC and NSCLC cell lines were plated on 96-well plates and treated with 25 $\mu$M compound **2** alone, 500 $\mu$M NAC alone or together for 3 d. Complete growth medium with drugs was replaced every 24 h. Cell proliferation was analyzed using CyQUANT proliferation assay. The graphs show the mean cell number ± SEM from three independent experiments relative to that for control cells (DMSO-treated). **(C, D, E, F)** Significant differences between control (DMSO-treated) and drug-treated cells were assessed by one-way ANOVA tests for (C, E, F), and bootstrap tests for (D) (*$P$ < 0.05, **$P$ < 0.01, ***$P$ < 0.001, #, not significant).

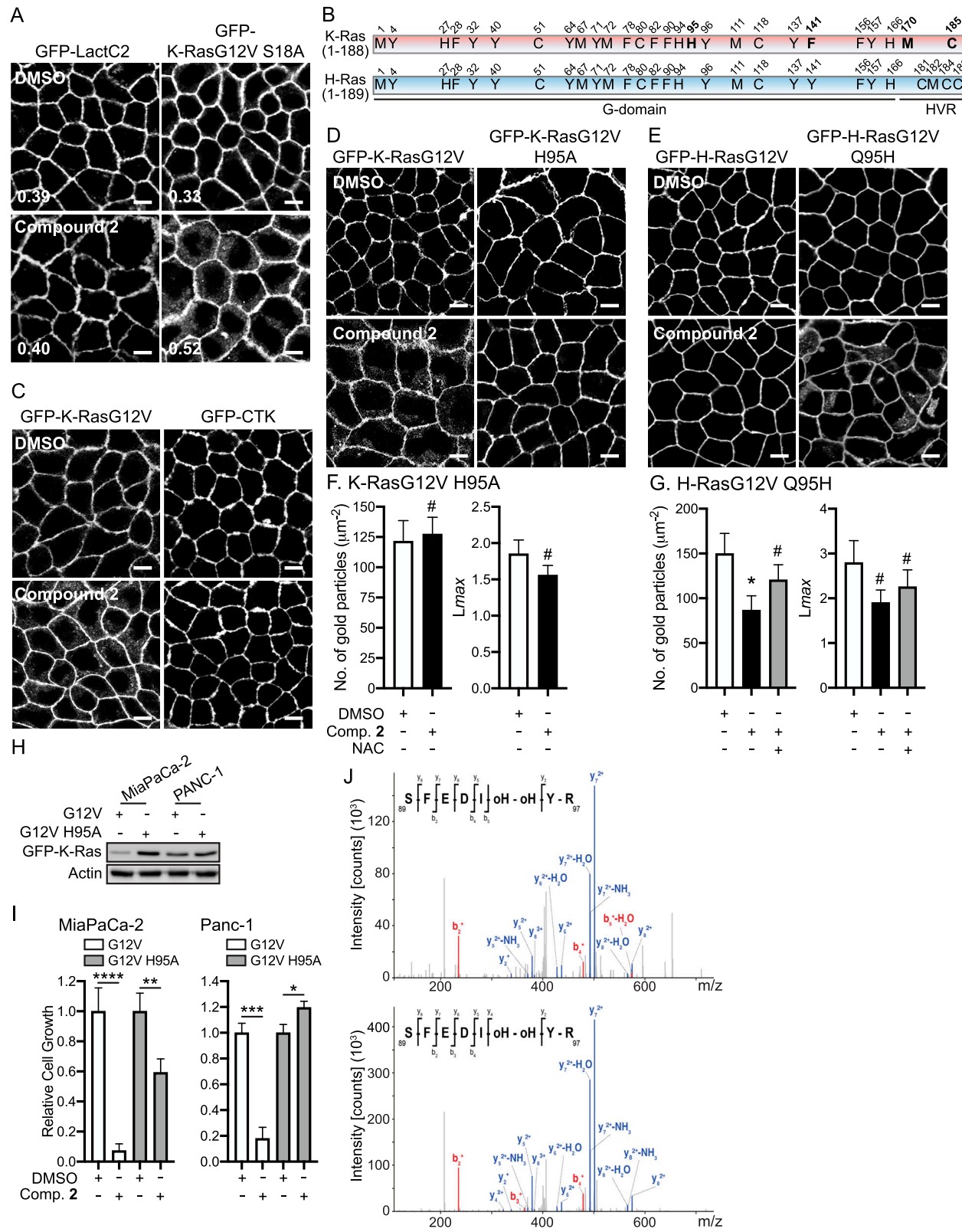

**Figure 5. K-Ras His95 residue may be oxidized, resulting in ROS-induced inhibition of K-Ras/PM binding and signaling.**
**(A)** MDCK cells co-expressing mCherry-CAAX and GFP-LactC2 or -K-RasG12V S181A were treated with 2 $\mu$M compound **2** for 48 h, and imaged by confocal microscopy. Inserted values represent an estimated mean fraction of mCherry-CAAX co-localized with GFP-tagged proteins calculated by Manders' coefficient from three independent experiments. Scale bar: 10 $\mu$m. Representative images are shown. **(B)** A schematic diagram of K- and H-Ras amino acid residues that can be oxidatively modified. The

from the PM (Fig 5A), suggesting that compound **2**-induced K-Ras/PM dissociation is independent of PM PtdSer abundance and of K-Ras Ser181 phosphorylation.

### K-Ras His95 is critical for compound 2-induced K-Ras/PM dissociation

Although H-Ras is oxidatively modified perturbing its activity, little is known about K-Ras oxidative modifications (Heo & Campbell, 2005; Heo et al, 2005; Burgoyne et al, 2012; Messina et al, 2019). Because our data demonstrate that cellular ROS elevation disrupts the PM binding of K-Ras, but not H-Ras, which is reversed by NAC supplementation, we hypothesized that ROS oxidatively modifies K-Ras, blocking K-Ras/PM binding and signaling. There are five amino acid residues that can be oxidatively modified and are unique to K-Ras (bold in Fig 5B). To identify the exact amino acid residue regulated by cellular ROS, we first generated MDCK cells stably expressing GFP-tagged C-terminal hypervariable region of K-Ras (GFP-CTK), sufficient for K-Ras transport to and stable interaction with the PM (Apolloni et al, 2000), and imaged its cellular localization after compound **2** treatment. We found that the compound had no effect on the PM localization of GFP-CTK (Fig 5C), suggesting that the target residue must be located within the G-domain. In the G-domain, K-Ras contains His at the 95th position, which can be oxidatively modified (this position is Gln in H-Ras, which does not get oxidatively modified). His95 is solvent-exposed in the crystal structure of K-Ras (Fig S6), and previously reported molecular dynamic simulations suggest His95 would still be accessible to solvent, and hence to ROS when K-Ras is bound to a model membrane (Prakash et al, 2016, 2019). To examine if His95 is involved in the ROS-induced K-Ras/PM dissociation, we substituted His with Ala, which does not undergo oxidative modification. MDCK cells expressing GFP-K-RasG12V H95A were treated with compound **2**, and K-Ras cellular localization was studied by confocal and electron microscopy. The compound had no effect on the PM binding and nanoclustering of K-RasG12V H95A (Fig 5D and F). We further substituted the Gln95 of H-Ras to His and studied its cellular localization after compound **2** treatment. Whereas Gln95His mutation did not alter the PM binding and nanoclustering of H-RasG12V under basal conditions (Fig S7), compound **2** treatment abrogated the PM binding and nanoclustering of H-RasG12V Q95H, and this effect was reversed by NAC

supplementation (Fig 5E and G). Moreover, to evaluate the role of K-Ras His95 residue on cancer growth, K-Ras-dependent PDAC cell lines ectopically expressing GFP-K-RasG12V or -K-RasG12V H95A were treated with compound **2** for 3 d, and their proliferation was measured. Our data show that the growth of cells expressing GFP-K-RasG12V H95A was less sensitive to compound **2**, compared with K-RasG12V–expressing cells (Fig 5H and I). These data suggest that the His95 residue is important for the ROS-induced K-Ras/PM dissociation and growth inhibition of K-Ras-dependent cancers.

To directly examine if the His95 residue is oxidized by compound **2**, GFP-K-RasG12V was immunopurified from MDCK cells expressing GFP-K-RasG12V and treated with compound **2** or $H_2O_2$ for 48 h. Liquid chromatography mass spectrometry (LC-MS/MS) analysis was performed and $MS^n$ fragmentation spectra suggest that although the peptide matching K-Ras His95 oxidation was detected, the fragments corresponding to the oxidized His residues were not detected. It is hypothesized to be a result of known issues with His oxidation and collision-induced dissociation fragmentation (Srikanth et al, 2009). Although the fragments were not directly detected, these data provide indirect evidence for oxidation at K-Ras His95 after compound **2** and $H_2O_2$ treatment (Fig 5J). Because His94 and His95 are the only His residues within the peptide (Fig 5B), and the PM binding of K-Ras His95Ala mutant is not changed by compound **2** (Fig 5D and F), our data support the induction of oxidation at K-Ras His95 by cellular ROS. Together, our data suggest that cellular ROS regulates K-Ras/PM binding, nanoclustering, signaling, and the growth of K-Ras-dependent cancer growth by oxidation at K-Ras His95 residue.

## Discussion

In this study, we have demonstrated that our novel ferrocene derivative, compound **2** ($C_{16}H_{20}FeClNO$) translocates K-Ras from the PM to cytosol and endomembranes by elevating cellular ROS, and it blocks the K-Ras/MAPK signaling and the growth of K-Ras-dependent PDAC and NSCLC cells. These effects were reversed by supplementation of NAC, a general antioxidant. Moreover, NAC alone enhances K-Ras/PM binding and K-Ras signaling without altering its PM nanoclustering. Although previous studies have reported the roles of K-Ras signaling on regulating redox balance,

---

residues unique to K-Ras are in bold. **(C, D, E)** MDCK cells stably expressing GFP-CTK (C), GFP-K-RasG12V or -K-RasG12V H95A (D), and GFP-H-RasG12V or -H-RasG12V Q95H (E) were treated with 2 $\mu$M compound **2** for 48 h, and imaged by confocal microscopy. Scale bar: 10 $\mu$m. Representative images from three independent experiments are shown. (Left panels in (**F, G**)) basal PM sheets prepared from MDCK cells expressing GFP-K-RasG12V, -K-RasG12V H95A, -H-RasG12V or -H-RasG12V Q95H and treated with 2 $\mu$M compound **2** with or without 500 $\mu$M N-acetylcysteine for 48 h were labeled with anti-GFP-conjugated gold and visualized by EM. The graphs show a mean number of gold particles ± SEM (n ≥ 20). (Right panels in (**F, G**)) Spatial mapping of the same gold-labeled PM sheets was performed. The peak values, $L_{max}$, of the respective weighted mean K-function $L(r)−r$ curves are shown as bar graphs (n ≥ 20). Significant differences between control (DMSO-treated) and compound **2**-treated cells were assessed by t tests for the no. of gold particles and bootstrap tests for L$max$ (*$P < 0.05$, #, not significant). **(H)** Cell lysates from K-Ras-dependent PDAC cell lines infected with lentivirus expressing GFP-K-RasG12V or -K-RasG12V H95A were immunoblotted using an anti-GFP antibody to measure GFP-K-Ras expression levels. Representative blots from three independent experiments are shown. Actin blots were used as loading controls. **(I)** These cells were seeded on a 96-well plate and treated with 25 $\mu$M compound **2** for 3 d. Complete growth medium containing the compound was replaced every 24 h. Cell proliferation was analyzed using CyQUANT proliferation assay. The graphs show the mean cell number ± SEM relative to that for control cells (DMSO-treated) from three independent experiments. Significant differences between control (DMSO-treated) and drug-treated cells were assessed by t tests (*$P < 0.05$, **$P < 0.01$, ***$P < 0.001$, ****$P < 0.0001$). **(J)** GFP-K-RasG12V was purified using GFP-Trap agarose beads from MDCK cells stably expressing GFP-K-RasG12V and treated with 2 $\mu$M compound **2** or 50 $\mu$M $H_2O_2$ for 48 h. The isolated protein was digested and subjected to liquid chromatography mass spectrometry analysis. The spectra represent the fragmentation of the K-Ras peptide (amino acid residues 89–97) from cells treated with compound **2** (upper panel) and $H_2O_2$ (lower panel). Fragmentation generates C′−N′ (b-ions in red) and N′−C′ (y-ions in blue). The differences between peaks correspond to amino acids of the sequence. All of the labels are the fragment matches of peptide.

our study reveals direct effects of the redox balance on K-Ras/PM binding and its signal output, thus providing a mechanistic link between K-Ras and recent reports on the antioxidant-induced growth and metastasis of K-Ras-driven NSCLC (Sayin et al, 2014; Lignitto et al, 2019; Wiel et al, 2019). Under oxidative stress, BACH1 is degraded by free heme released by the oxidation of heme-containing proteins (Zenke-Kawasaki et al, 2007; Lignitto et al, 2019). Prolonged supplementation of antioxidants reduces cellular ROS levels, which lowers cellular free heme, stabilizing BACH1 protein. The elevated BACH1 further promotes glycolysis by up-regulating transcription of *GAPDH* and *hexokinase-2*, promoting metastasis in K-Ras-driven NSCLC (Wiel et al, 2019) (Fig 6). Consistently, promoting the endogenous antioxidant program either by activating NRF2 (nuclear factor erythroid-derived 2-like 2), the master transcriptional regulator of antioxidant responses, or blocking its negative regulator, KEAP1 (Kelch-like ECH-associated protein 1), up-regulates heme oxygenase-1, which degrade free heme, resulting in the BACH1-dependent metastasis of K-Ras-driven NSCLC (Li & Stocker, 2009; Wiel et al, 2019). Moreover, prolonged treatment of antioxidants, NAC or vitamin E stimulates the growth of K-Ras-driven NSCLC via disrupting the ROS–p53 axis (Sayin et al, 2014). Thus, these studies have demonstrated the role of antioxidants in the ROS/p53-regulated growth of and the BACH1/glycolysis-mediated metastasis of K-Ras-driven NSCLC, and our study adds the pivotal role of oncogenic K-Ras in this signal axis. We have showed that compound **2** promotes BACH1 degradation in K-Ras-dependent PDAC and NSCLC cell lines, and NAC alone promotes K-Ras/PM binding and ERK phosphorylation in a model cell line. We further provided indirect evidence that cellular ROS oxidatively modifies K-Ras at His95 residue, and the growth of cancer cells ectopically expressing K-RasG12V H95A were resistant to compound **2**. Together, we propose that in K-Ras-driven NSCLC and possibly PDAC, elevation of cellular antioxidants prevents the oxidative modification of the His95 residue of oncogenic mutant K-Ras, promoting stable K-Ras/PM binding, and thereby the MAPK signaling, which results in elevated cell proliferation. This elevated K-Ras signaling further promotes NRF2 expression, stimulating the NRF2-mediated antioxidant program (Singh et al, 2008; DeNicola et al, 2011; Yang et al, 2021). This in turn, provides a positive-feedforward loop for the antioxidant/BACH1/glycolysis-induced metastatic signaling. The enhanced K-Ras signaling also promotes glycolysis and metastasis pathways (Ying et al, 2012; Cox et al, 2014; Gorfe & Cho, 2019), further stimulating the antioxidant/BACH1/glycolysis-induced NSCLC metastasis (Fig 6). However, one of caveat with our model is that NAC treatment for 3 d did not stimulate the growth of K-Ras-dependent cancer cells (Fig 4F). We speculate that NAC treatment for 3 d may be insufficient timeframe to reprogram signal network for translating NAC-promoted K-Ras/PM binding and K-Ras signal output to cell proliferation in human cancer cells. For example, K-RasG12D-expressing primary fibroblasts were treated with NAC for 12 d to show elevated cell proliferation (Sayin et al, 2014).

We have identified K-Ras His95 as a novel amino acid residue for regulating K-Ras/PM binding via oxidative modification. Although the exact mechanisms on how oxidative modification of K-Ras His95 perturbs the PM binding needs to be further elucidated, the observations that ROS elevation does not affect the PM binding of

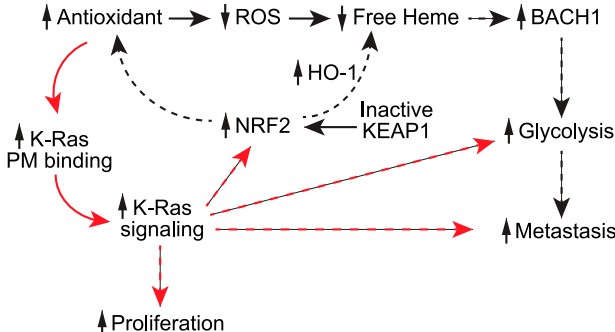

**Figure 6. Proposed model for the role of K-Ras in the antioxidant-induced growth and metastasis of K-Ras-driven cancers.**
Elevation of cellular antioxidant by prolonged antioxidant supplement or activating NRF2 blocks the oxidative modification of K-Ras His95 residue, enhancing K-Ras/PM binding, and thereby K-Ras signaling. This in turn stimulates Ras downstream effectors regulating proliferation, glycolysis and metastasis, providing a positive feedforward loop for the antioxidant/BACH1/glycolysis-induced metastasis of K-Ras-driven cancers. Solid and dotted lines indicate direct and indirect mechanisms, respectively. PM, plasma membrane; NFR2, nuclear factor erythroid-derived 2-like 2; BACH1, BTB and CNC homology 1; KEAP1, Kelch-like ECH-associated protein 1; HO-1, heme oxygenase-1.

K-Ras4A, which also has His95, and that Gln95His substitution dissociates H-Ras from the PM (Figs 2 and 5) provide a hint into the possible involvement of the PDE6δ-Arl2/3 transport machinery. PDE6δ continuously sequesters K-Ras and depalmitoylated H- and N-Ras from endomembranes via interacting with their C-terminal farnesyl moiety. The release factors Arl2 and 3 then bind the Ras/PDE6δ complex and release Ras to perinuclear membranes for returning to the PM (Ismail et al, 2011; Chandra et al, 2012; Schmick et al, 2014). Although PDE6δ does not bind depalmitoylated K-Ras4A likely because of steric hindrance caused by two Lys residues immediately before the C-terminal farnesylated Cys of K-Ras4A, PDE6δ directly interacts with five amino acids (residues 180–184) immediately before the farnesylated Cys of K-Ras for the stable binding (Dharmaiah et al, 2016). Thus, it is plausible that oxidatively modified His95 of K-Ras may perturb Arl2/3 interaction with the K-Ras/PDE6δ complex, resulting in K-Ras distribution from the PM to endomembranes.

In conclusion, our study demonstrates that redox system directly regulates K-Ras/PM binding and K-Ras signal output via oxidative modification at His95, and proposes a role of oncogenic mutant K-Ras in the antioxidant-induced growth and metastasis of K-Ras-driven cancers.

# Materials and Methods

### Synthesis of 1-ferrocenyl-2-[(dimethylamino)methyl]-2-propen-1-one (compound 1)

All synthetic manipulations were done under a nitrogen atmosphere unless otherwise noted. All glasswares were oven-dried at 110°C for 12 h before use. Acetylferrocene was prepared according to the literature procedure (Graham et al, 1957; Donahue & Donahue, 2013). Bis(dimethylamino)methane, phosphoric acid,

acetic acid, and hydrochloric acid (1 M) in ether were purchased from commercial sources and used as received. Solvents were dried with a solvent purification system from Inert Pure Company (THF, $CH_2Cl_2$, $Et_2O$, and toluene) and degassed using three freeze–pump–thaw cycles before use. All solvents were stored over 4 Å molecular sieves in a nitrogen-filled glove box. $CDCl_3$ (99.9%) and $D_2O$ (99.9%) were purchased from Acros Laboratories and dried over 4 Å molecular sieves before use. [1]H and [13]C NMR spectra were recorded on a Bruker 300 MHz NMR spectrometer. Spectra were referenced to the residual solvent as an internal standard for [1]H NMR: $CDCl_3$, 7.26 ppm, $D_2O$ 4.79 ppm; for [13]C NMR: $CDCl_3$, 77.16. Coupling constants (J) are expressed in hertz (Hz). Elemental analyses were performed by Midwest Microlab, LLC. Acetylferrocene (1.39g, 6.1 mmol) was added to a premixed solution of bis(dimethylamino)methane (1.47 ml, 1.77 mmol), phosphoric acid (0.65 ml, 9.5 mmol), and 14 ml of acetic acid in a two-neck round-bottom flask equipped with a stir bar and condenser. Under a nitrogen atmosphere, the mixture was refluxed at 110°C and stirred for 5 h. The mixture was allowed to cool to RT, diluted with 14 ml of water, and extracted two times with 50 ml of diethyl ether. The aqueous solution was then cooled using an ice–water mixture and made alkaline by adding solid NaOH pellets (6g) and extracted three times with 50 ml of diethyl ether. Finally, the organic solution was washed with water, dried in $MgSO_4$, and concentrated under reduced pressure. The crude residue was passed through short silica gel column chromatography using (Hexane:DCM:$Et_3N$, 80:10:10) and yielded 1-ferrocenyl-2-[(dimethylamino)methyl]-2-propen-1-one. 1.29g, 71% yield. [1]H NMR ($CDCl_3$, 300 MHz): d 5.86 (s, 1H), 5.69 (s, 1H), 4.84 (s, 2H), 4.51 (s, 2H), 4.24 (s, 5H), 3.25 (s, 2H), 2.28 (s, 6H). [13]C NMR ($CDCl_3$, 75 MHz): d 200.1, 146.9, 121.7, 77.9, 72.2, 70.9, 70.1, 61.6, 45.5.

### Synthesis of 1-ferrocenyl-2-[(dimethylamino)methyl]-2-propen-1-one hydrochloride (compound 2)

Compound **1** (1.19g, 4.0 mmol) and dichloromethane (20 ml) were charged into a 40-ml glass vial, and the reaction mixture was cooled using an ice bath. Hydrochloric acid (1 M solution in diethyl ether, 4 ml, 4.0 mmol) was added dropwise. Reaction contents were stirred for 1 h, and the solvent was removed under vacuum; the residue was washed with diethyl ether (2 × 10 ml) and dried under vacuum, yielding compound **2**, 1.27g, 95% yield. [1]H NMR ($D_2O$, 300 MHz): d 6.73 (s, 1H), 6.37 (s, 1H), 4.93-4.84 (m, 4 H), 4.31 (s, 5 H), 3.98 (s, 2 H), 2.87 (s, 2H). [13]C NMR ($D_2O$, 75 MHz): d 200.3, 136.3, 134.98, 75.6, 74.63, 71.52, 70.85, 70.57, 59.47, 42.64. HRMS (ESI) for $[C_{16}H_{20}FeNO]^+[M]^+$ Calcd. 298.0889 found. 298.0884. Anal. Calcd. For: $C_{16}H_{20}ClFeNO$: C, 57.60, H, 6.04, N, 4.20; found: C, 56.87, H, 5.89, N, 4.21.

### Plasmids and reagents

The following antibodies to measure Ras signaling were purchased from Cell Signaling Technology: pAkt (Ser473) (Cat# 4060), total Akt (Cat# 2920), ppERK (Thr202/Tyr204) (Cat# 4370), total ERK (Cat# 4696), p-p90RSK (Cat #9344), and total RSK (Cat #9355). The following antibodies used to measure housekeeping genes were purchased from Proteintech: β-actin (Cat# 66009-1-Ig), GFP-tag (Cat# 60002-1-Ig), α-tubulin (Cat# 11224-1-AP), BACH1 (Cat# 66762-1-IG), H-Ras (Cat# 18295-1-AP), K-Ras2B (Cat# 16155-1-AP), and K-Ras2A (Cat #16156-1).

N-Ras antibody (F155) (Cat #sc-31) was purchased from Santa Cruz Biotechnology. The RFP-tagged organelle markers were purchased from Invitrogen: CellLight Golgi-RFP BacMam 2.0 (Cat# C10593), CellLight Lysosomes-RFP BacMam 2.0 (Cat #C10597), CellLight Mitochondria-RFP BacMam 2.0 (Cat #C10601), CellLight Late Endosome-RFP BacMam 2.0 (Cat #C10589), CellLight Early Endosome-RFP BacMam 2.0 (Cat #C10587), and ER-Tracker Red (Cat #E34250; BODIPY TR Glibenclamide). Mitomycin C was purchased from Alfa Aesar Co. (Cat #J63193). Carmustine was purchased from MedChemExpres (Cat # HY-13585/CS-2935). N-acetylcysteine was purchased from Sigma-Aldrich (Cat # A7250-10G). Hydrogen peroxide ($H_2O_2$) was purchased from Sigma-Aldrich (Cat #H1009).

### Cell culture

MDCK were maintained in DMEM; Cat #10569-010; Gibco. Human PDACs cells: BxPC3, Panc 10.05, and AsPC-1 were maintained in RPMI-1640 (30-2001; ATCC), MiaPaCa2 and PANC-1 were maintained in DMEM, HPAC were maintained in DMEM-F12 (Cat # 11320033; Gibco), and HPAF-II were maintained in EMEM (30-2003; ATTC). Human NSCLCs: H522, H1975, H1299, A549, H23, H441, H1703, and H358 were maintained in RPMI-1640. All cancer cell lines were maintained in media supplemented with 10% FBS (Cat #16000-069; Gibco) and 2 mM L-glutamine (Cat # CA009-010; GenDEPOT). Cells were test for mycoplasma (Cat# LT07-710; MycoAlert PLUS Mycoplasma Detection Kit). All cell lines were maintained in an incubator at 37°C at 5% $CO_2$.

### Proliferation assay

PDAC cells were seeded at 3 × 10[5] and NSCLC cells were seeded at 1 × 10[5] – 2 × 10[5] on a 96-well plate. 24 h later, cells were incubated with DMSO (control) or increasing concentrations of $C_{16}H_{20}FeClNO$ in 100 µl complete growth media for 72 h, changing the media every 24 h. Cell proliferation was quantified using CyQuant NF Cell Proliferation Assay Kit (Cat # 35006; Molecule Probes) according to the manufacturer's protocol. Plates were read using BioTek Synergy H1 microplate reader at excitation/emission of 480/530 nm.

### Confocal microscopy

MDCK cells were seeded on coverslips at 2.5 × 10[5] on a 12-well plate. Cells were treated with increasing concentrations of $C_{16}H_{20}FeClNO$, mitomycin C, carmustine or $H_2O_2$ for 48 h. Cells were washed twice with ice-cold 1x PBS and then fixed with 4% PFA (Cat #15710; Electron Microscopy Services) followed by 50 mM $NH_4Cl$. Slides were imaged using the Olympus FV1000 confocal microscope using a 60x objective. Images were quantified for co-localization using Manders' coefficient on ImageJ software (version 1.52a).

### Electron microscopy (EM) and spatial mapping

Plasma membrane sheets were prepared and fixed as previously described (Prior et al, 2003; Hancock & Prior, 2005; Garrido et al, 2020). For univariate analysis, plasma membrane sheets were labeled with anti-GFP antibody conjugated to 4.5-nm gold particles. Digital images of the immunogold-labeled plasma membrane

sheets were taken in a transmission electron microscope. Intact 1 $\mu m^2$ areas of the plasma membrane sheet were identified using ImageJ software, and the $(x, y)$ coordinates of the gold particles were determined (Prior et al, 2003; Hancock & Prior, 2005). K-functions (Ripley, 1977) were calculated and standardized on the 99% confidence interval (CI) for univariate functions (Diggle et al, 2000; Prior et al, 2003; Hancock & Prior, 2005). In the case of univariate functions, a value of $L(r)—r$ greater than the CI indicates significant clustering, and the maximum value of the function ($L_{max}$) estimates the extent of clustering. Differences between replicated point patterns were analyzed by constructing bootstrap tests as described previously (Diggle et al, 2000; Plowman et al, 2005), and the statistical significance against the results obtained with 1,000 bootstrap samples was evaluated.

### Subcellular fractionation assay

MDCK cells were seeded at $1.7 \times 10^6$ onto 10 cm dishes. Then, cells were treated with increasing concentrations of $C_{16}H_{20}FeClNO$, mitomycin C, carmustine or $H_2O_2$ for 48 h. Dishes were washed twice with ice-cold 1x PBS. Lysates were harvested in buffer A containing 10 mM Tris pH 7.5, 75 mM NaCl, 25 mM NaF, 5 mM $MgCl_2$, 1 mM EGTA, 1 M DTT, and 100 $\mu M$ $NaVO_4$. Lysates were spun at 100,000$g$ for 1 h at 4°C using Sorvall Discovery MX120SE Ultracentrifuge (Thermo Fisher Scientific) to isolate the cytosolic fraction. Pellets were resuspended in Buffer A and sonicated 10x at 4°C to isolate the membrane-bound fraction. Protein concentrations were determined using the BCA protein assay (Reagent A Cat #23228; Reagent B Cat # 1859078; Thermo Fisher Scientific). SDS–PAGE and immunoblotting were performed using 20–25 $\mu g$ of lysate from each sample group. Blots were detected using chemiluminescence on the Amersham Imager 600 (GE Healthcare Life Sciences). Blots were quantified using ImageJ software.

### Western blotting

MDCK cells were seeded at $3 \times 10^5$ on a six-well plate and treated with increasing concentrations of $C_{16}H_{20}FeClNO$ for 48 h. Then, cells were washed twice with ice-cold 1x PBS. Cells were harvested in a lysis buffer consisting of 50 mM Tris pH 7.5, 25 mM NaF, 5 mM $MgCl_2$, 75 mM NaCl, 5 mM EGTA, 1 mM DTT, 100 $\mu M$ $NaVO_4$, and 1% NP-40. Protein concentrations were determined using the BCA protein assay. SDS–PAGE and immunoblotting were performed using 20–25 $\mu g$ of lysate from each sample group. Blots were detected using chemiluminescence on the Amersham Imager 600. Blots were quantified using ImageJ software.

### Quantification of ROS levels by $H_2DCFDA$

WT MDCK cells were seeded at $2 \times 10^4$ on a clear-bottomed 96-well plate overnight. Then, cells were co-treated with 25 $\mu M$ of DCFDA dye ($H_2DCFDA$; Cat #D399; Thermo Fisher Scientific) and either $H_2O_2$, $C_{16}H_{20}FeClNO$ only or $C_{16}H_{20}FeClNO$ with NAC for 48 h. After 48 h, fluorescence was read using the BioTek Synergy H1 microplate reader with emission/excitation at 495/527 nm.

### In-gel digest with LC-MS/MS analysis

For the digestion, each gel band was washed and destained in 50% methanol (Optima LC/MS, A546; Thermo Fisher Scientific)/5% acetic acid (27225; Millipore Sigma) for a minimum of 2 h, followed by dehydration in acetonitrile. The gel pieces were rehydrated with 10 mM dithiothreitol (D0652; Millipore Sigma) in 0.1 M ammonium bicarbonate and reduced at RT for 0.5 h. The DTT solution was removed and the sample was alkylated with 50 mM iodoacetamide (I6125; Millipore Sigma) in 0.1 M ammonium bicarbonate (A6141; Millipore Sigma) at RT for 0.5 h in the dark. The iodoacetamide reagent was removed and the gel pieces dehydrated in acetonitrile before being rehydrated and washed with 100 mM ammonium bicarbonate, and dehydrated in acetonitrile. The protease (Trypsin/Lys-C, V5073; Promega) was driven into the gel pieces by rehydrating them in 20 ng/ml trypsin/Lys-C in 50 mM ammonium bicarbonate on ice for 10 min. Excess trypsin solution was removed and 50 mM ammonium bicarbonate added before overnight digestion at RT. The peptides formed were extracted from the polyacrylamide with two washes with 50% acetonitrile/ 5% formic acid (85178; Thermo Fisher Scientific). These extracts were combined and evaporated to ~25 $\mu l$ for analysis by LC-MS/MS.

6 $\mu l$ of in-gel samples were separated on an RSLCnano (Thermo Fisher Scientific). Peptides were preconcentrated on a PepMap C18 5, 300 $\mu m \times 5$ mm trap column with 5 $\mu l$/min isocratic flow of 2% acetonitrile:0.03% trifluoroacetic acid (aq) for 7.5 min. Analytical reverse phase separations were performed on a PepMap C18 3 $\mu m$, 100 Å, 75 $\mu m \times 150$ mm operated at 35°C with a 300 nl/ min flow. Mobile phases A consisted of 0.1% formic acid (aq) and B consisted of 0.1% formic acid in acetonitrile (Optima MS Grade; Thermo Fisher Scientific). The analytical gradient was 2% B from 0–10 min, 30% B at 46 min, 40% B at 48 min, 90% B at 50.5 min with a 4 min hold. The separation returned to starting conditions at 56.5 min and held there until 65 min. Profile spectra of eluted peptides were detected on an Orbitrap Fusion Lumos mass spectrometer operated in positive ion mode affixed with an EasySpray ion source with a spray voltage of 2.3 kV and an ion transfer temperature of 275°C. MS1 scans were collected every 3 s at 120,000 resolution across 375–2,000 m/z with an AGC of $4 \times 10^5$ ions or 50 ms time. Data-dependent $MS^n$ scans were collected in the Orbitrap with precursors isolated at 1.2 m/z in the quadrupole. Collision induced dissociation was performed in the ion trap at 35% normalized collision energy over 10 ms. The Orbitrap was operated at 30,000 resolution with AGC settings of $5 \times 10^4$ using dynamic injection time collecting two microscan profile scans.

Data were searched using the SequestHT function against the human K-Ras protein (UniProt ID P01116) within the Proteome Discoverer Software suite (v. 2.5; Thermo Fisher Scientific). Search parameters were as follows: MS(n-1) precursors, trypsin full specificity, 4 missed cleavages, 10 ppm MS1 tolerance, 0.02 kD $MS^n$ tolerance, oxidation of His and Met, carbamidomethyl modification of cysteine, and acetyl N-terminal modification. The peptide spectral match validator was used using strict false discovery for both peptides and protein identifications. Spectra matching the peptide of interest were exported.

## Statistics

GraphPad prism software (v.8.4.0) was used for one-way ANOVA, $t$ test, and generating response curve graphs and $IC_{50}$s.

# Supplementary Information

# Acknowledgements

This work was supported by NIH (R01 GM144836) to AA Gorfe, Wright State Seed Grant Program to K-J Cho, and NIH (R15 CA232765), The American Chemical Society Petroleum Research Fund (PRF-59893-UR7), and Wright State University CoSM fund to K Arumugam.

## Author Contributions

KM Rehl: conceptualization, data curation, formal analysis, validation, investigation, methodology, and writing—original draft, review, and editing.
J Selvakumar: resources.
RL Pitsch: investigation and writing—original draft, review, and editing.
D Hoang: resources.
K Arumugam: resources, formal analysis, and writing—original draft.
SW Harshman: data curation, formal analysis, and writing—original draft, review, and editing.
AA Gorfe: formal analysis and writing—original draft, review, and editing.
K-J Cho: conceptualization, data curation, formal analysis, supervision, funding acquisition, validation, investigation, methodology, and writing—original draft, review, and editing.

## Conflict of Interest Statement

The authors declare no competing interests. The views expressed are those of the authors and do not reflect the official guidance or position of the United States Government, the Department of Defense or of the United States Air Force.

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
