## [Reviewer comments · Life Science Alliance]

Life Science Alliance

A new ferrocene derivative blocks K-Ras localization and function by oxidative modification at His95

Kristen Rehl, Jayaraman Selvakumar, Rhonda Pitsch, Don Hoang, Kuppuswamy Arumugam, Sean Harshman, Alemaheyu Gorfe, and Kwang-Jin Cho

DOI: <https://doi.org/10.26508/lsa.202302094>

Corresponding author(s): Kwang-Jin Cho, Boonshoft School of Medicine, Wright State University, Dayton

Review Timeline:

Submission Date:	2023-04-12
Editorial Decision:	2023-05-15
Revision Received:	2023-08-14
Editorial Decision:	2023-08-21
Revision Received:	2023-08-23
Accepted:	2023-08-24

Scientific Editor: Novella Guidi

Transaction Report:

May 15, 2023

Re: Life Science Alliance manuscript #LSA-2023-02094-T

Dr. Kwang-Jin Cho
Wright State University
3640 Colonel Glenn Hwy
Dayton 45435

Dear Dr. Cho,

Thank you for submitting your manuscript entitled "A new ferrocene derivative blocks KRAS localization and function by oxidative modification at His95." to Life Science Alliance. The manuscript was assessed by expert reviewers, whose comments are appended to this letter. We invite you to submit a revised manuscript addressing the Reviewer comments.

Thank you for this interesting contribution to Life Science Alliance. We are looking forward to receiving your revised manuscript.

Sincerely,

B. MANUSCRIPT ORGANIZATION AND FORMATTING:

Reviewer #1 (Comments to the Authors (Required)):

The manuscript by Rehl and colleagues describes a novel ferrocene compound that indirectly prevents localization of KRAS4B to the plasma membrane via ROS-mediated oxidative effects. The authors then go on to show that histidine 95 in KRAS4B could be responsible for PM localization by demonstrating that replacing H95 with other amino acids prevents loss of PM localization. They argue that the activity of the novel ferrocene compound depends on oxidative stress since it can be rescued with antioxidants. Moreover, other inducers of oxidative stress also cause redistribution of KRAS4B. The manuscript is well written and has some interesting findings. However, as indicated below, some parts of the manuscript require further substantiation.

1. Is PM localization of endogenous KRAS4B in the cell lines used in Fig. 1B also affected? Here, the authors could perform simple fractionation experiments as they did in Fig 2F. One would expect that only KRAS4B, but not 4A is excluded from the PM upon treatment with the ferrocene compound.
2. It is not entirely clear whether the cytotoxic effect of the ferrocene compound is really largely caused by oxidative stress (affecting a variety of mechanisms) or specifically depends on KRAS4B mislocalization. Can the cytotoxic effect be rescued in ferrocene-sensitive NSCLC/PDAC cell lines with a KRAS mutant that carries the H95 mutation? On the other hand, selective depletion of KRAS4B should then give the same result as observed in Fig. 1B.
3. Moreover, the authors should test whether the cytotoxic effects the ferrocene compound (Fig. 1B) can be rescued with NAC.
4. It is also important to show how KRAS signaling (pERK, pAKT) is affected in cell lines that are sensitive to the compound, assuming these are KRAS-dependent cell lines.
5. The long discussion about the potential link between KRAS signaling and Bach1 stabilization is not really substantiated by data. While these two activities could well be related, no real evidence is presented (for such a long discussion including a scheme in Fig. 5H). Even more, stabilization of Bach1 has been linked with metastasis whereas the experiments presented in this manuscript do not touch upon this. First, treatment with the ferrocene compound should decrease Bach1 levels. In addition, if this link was relevant, depletion of Bach1 should cause the same effect on cell growth as the compound. Yet, the entire Bach1 network has only been linked with metastasis so that it is possible that Bach1 depletion has no effect at all on cell proliferation.

Reviewer #2 (Comments to the Authors (Required)):

This is a very interesting study. The authors developed a ferrocene compound targeting the K-Ras mutant and investigated the mechanism of its K-Ras-dependent anti-PDACH activity and anti-NSCLChs activity.

The importance of K-Ras mutants as targets in the context of the current difficulties in the development of targeted small molecule chemical drugs. The authors' manuscript is very interesting, i.e., targeting PM binding and signaling of K-Ras through the properties of ferrocene derivatives that enhance ROS. Finally, the His95 residue, which plays a key role, is also targeted. The authors have done a very good job throughout the manuscript and all experimental results are demonstrated by robust experiments.

At present, I have the following questions:

The authors do not mention how the molecules were designed (which is a bit confusing compared to the detailed biological experiments that follow). Of course, it is perfectly acceptable that the molecules were designed from "screening" or "experimentation". However, if the authors are interested, I would suggest providing some information on how the authors designed the molecular structure.

2. The authors provide a lot of data on drug activity, which is very interesting. But were any toxicological studies performed?

Reviewer #3 (Comments to the Authors (Required)):

Ferrocene derivatives are known to increase cellular reactive oxygen species (ROS) and to thereby have anti-proliferative

effects on some cancer cells. Rehl et al. synthesized a novel ferrocene compound, C₁₆H₂₀FeCINO (FeCINO), and found that it inhibits the proliferation of KRAS-mutant tumor cells. Using a variety of methods, the authors show that treatment of cells with FeCINO reduces the pool of KRAS, but not HRAS or NRAS, on the plasma membrane (PM), diminishes nanoclustering, and reduces MAPK signaling. The effects of FeCINO were ascribed to elevated ROS because other agents known to increase ROS (e.g. mitomycin C) had similar effects and because N-acetylcysteine (NAC) reversed the effects. Having ruled out effects on KRAS phosphorylation and PM phosphatidylserine content known to affect PM association of KRAS, the authors entertained the hypothesis that KRAS is directly modified by ROS. They focused on H95 because they mapped the effect to the G domain and noted that H95 is a solvent-exposed, oxidation-susceptible residue in KRAS that does not occur in HRAS. PM association and nanoclustering of an H95A mutant was unaffected by FeCINO and a reciprocal Q95H substitution in HRAS rendered it sensitive. The authors conclude that FeCINO inhibits KRAS membrane association and signaling by oxidation of H95.

The study is well designed, well performed and clearly presented. The findings are of interest to RAS biologists and may have translational potential. The major weakness is that the authors do not provide direct evidence of FeCINO-induced oxidation of H95. The mutational analysis is indirect and does not prove oxidation. However, obtaining direct evidence of histidine oxidation by mass spectroscopy can be challenging (PMID: 19160434). If oxidized histidine can be detected, determination of the stoichiometry of modification will be important to explain how the modification affects signaling.

Minor points:

On page 4 the authors write that "PDE6 δ binds endocytosed Ras proteins via the farnesyl moiety." Endocytosis is not part of the process through which PDE6 δ extracts farnesylated proteins from membranes.

The effects of FeCINO on the growth of KRAS mutant and wild-type cells shown in Fig. 1 should be shown as a dose response, at least for a subset of sensitive and insensitive cell lines.

Reviewer #1 (Comments to the Authors (Required)):

The manuscript by Rehl and colleagues describes a novel ferrocene compound that indirectly prevents localization of KRAS4B to the plasma membrane via ROS-mediated oxidative effects. The authors then go on to show that histidine 95 in KRAS4B could be responsible for PM localization by demonstrating that replacing H95 with other amino acids prevents loss of PM localization. They argue that the activity of the novel ferrocene compound depends on oxidative stress since it can be rescued with antioxidants. Moreover, other inducers of oxidative stress also cause redistribution of KRAS4B. The manuscript is well written and has some interesting findings. However, as indicated below, some parts of the manuscript require further substantiation.

1. Is PM localization of endogenous KRAS4B in the cell lines used in Fig. 1B also affected? Here, the authors could perform simple fractionation experiments as they did in Fig 2F. One would expect that only KRAS4B, but not 4A is excluded from the PM upon treatment with the ferrocene compound.

We thank R1 for the valuable suggestion. We have performed fractionation assay using wild-type PANC-1 cell line to measure membrane-associated and cytosolic endogenous Ras isoforms (K-Ras4B, K-Ras4A, H-Ras and N-Ras). We showed that only K-Ras4B is dissociated from the membrane after the drug treatment, which is shown in Fig. 2H and described in lines 207 – 209.

2. It is not entirely clear whether the cytotoxic effect of the ferrocene compound is really largely caused by oxidative stress (affecting a variety of mechanisms) or specifically depends on KRAS4B mislocalization. Can the cytotoxic effect be rescued in ferrocene-sensitive NSCLC/PDAC cell lines with a KRAS mutant that carries the H95 mutation? On the other hand, selective depletion of KRAS4B should then give the same result as observed in Fig. 1B.

We thank R1 for the important comment. We understand that the best experimental design to address this comment is to introduce a point mutation, using CRISPR/Cas9, at the His95 (H95 -> A) in endogenous K-Ras of ferrocene-sensitive cancer cell lines, and measure their growth after ferrocene treatment. However, we were not able to do so due to lack of time. Instead, we infected ferrocene-sensitive PDAC cell lines using lentivirus expressing GFP-K-RasG12V or -K-RasG12V H95A and treated them with ferrocene for 3 days. Our data (Figs. 5H and I) show that the growth of cell lines ectopically expressing GFP-K-RasG12V H95A were less sensitive to ferrocene than that for K-RasG12V, suggesting that the ferrocene-induced growth inhibition is, at least in part, via K-RasHis95 residue. This is discussed in lines 331-337.

3. Moreover, the authors should test whether the cytotoxic effects the ferrocene compound (Fig. 1B) can be rescued with NAC.

We performed the suggested experiments. Fig. 4F now shows that NAC supplementation rescues the growth inhibitory effect of ferrocene in cancer cells. This is discussed in lines 279-280.

4. It is also important to show how KRAS signaling (pERK, pAKT) is affected in cell lines that are sensitive to the compound, assuming these are KRAS-dependent cell lines.

We measured ppERK, pAkt and p-p90RSK in ferrocene-sensitive PDAC and NSCLC cell lines (Fig. 1F), which is discussed in lines 160 – 173.

5. The long discussion about the potential link between KRAS signaling and Bach1 stabilization is not really substantiated by data. While these two activities could well be related, no real evidence is presented (for such a long discussion including a scheme in Fig. 5H). Even more, stabilization of Bach1 has been linked with metastasis whereas the experiments presented in this manuscript do not touch upon this. First, treatment with the ferrocene compound should decrease Bach1 levels. In addition, if this link was relevant, depletion of Bach1 should cause the same effect on cell growth as the compound. Yet, the entire Bach1 network has only been linked with metastasis so that it is possible that Bach1 depletion has no effect at all on cell proliferation.

We thank R1 for the important and valuable comment and suggestion, which we agree with. To make our model stronger, we now include a study reporting prolonged treatment of antioxidant stimulates the growth of K-Ras-driven NSCLC via disrupting the ROS-p53 axis (Sayin et al., 2014, Sci Transl Med) in Discussion. Moreover, we now include data demonstrating that 1) our compound reduces endogenous BACH1 protein expression in human cancers (Fig. 3D), 2) NAC alone elevates K-Ras/PM binding without altering K-Ras nanoclustering (Figs. 4C and D) and 3) NAC alone increases ppERK in K-RasG12V-expressing model cell line (Fig. 4E). Together, we propose that prolonged NAC treatment promotes K-Ras/PM binding and K-Ras signaling, which ultimately stimulates cancer growth and the BACH1-mediated cancer metastasis.

Reviewer #2 (Comments to the Authors (Required)):

This is a very interesting study. The authors developed a ferrocene compound targeting the K-Ras mutant and investigated the mechanism of its K-Ras-dependent anti-PDACH activity and anti-NSCLCs activity.

The importance of K-Ras mutants as targets in the context of the current difficulties in the development of targeted small molecule chemical drugs. The authors' manuscript is very interesting, i.e., targeting PM binding and signaling of K-Ras through the properties of ferrocene derivatives that enhance ROS. Finally, the His95 residue, which plays a key role, is also targeted. The authors have done a very good job throughout the manuscript and all experimental results are demonstrated by robust experiments.

At present, I have the following questions:

The authors do not mention how the molecules were designed (which is a bit confusing compared to the detailed biological experiments that follow). Of course, it is perfectly acceptable that the molecules were designed from "screening" or "experimentation". However, if the authors are interested, I would suggest providing some information on how the authors

designed the molecular structure.

We thank R2 for the valuable suggestion, which has been included in the revised manuscript in lines 133 – 141.

2. The authors provide a lot of data on drug activity, which is very interesting. But were any toxicological studies performed?

The only toxicological studies we have performed was to examine cellular apoptosis after ferrocene-treatment by immunoblotting cleaved caspase e and PARP-1, which is shown in Fig. S4 and discussed in lines 236 – 239. For in vivo studies, we are planning to apply for a grant funding in near future.

Reviewer #3 (Comments to the Authors (Required)):

Ferrocene derivatives are known to increase cellular reactive oxygen species (ROS) and to thereby have anti-proliferative effects on some cancer cells. Rehl et al. synthesized a novel ferrocene compound, C₁₆H₂₀FeCINO (FeCINO), and found that it inhibits the proliferation of KRAS-mutant tumor cells. Using a variety of methods, the authors show that treatment of cells with FeCINO reduces the pool of KRAS, but not HRAS or NRAS, on the plasma membrane (PM), diminishes nanoclustering, and reduces MAPK signaling. The effects of FeCINO were ascribed to elevated ROS because other agents known to increase ROS (e.g. mitomycin C) had similar effects and because N-acetylcysteine (NAC) reversed the effects. Having ruled out effects on KRAS phosphorylation and PM phosphatidylserine content known to affect PM association of KRAS, the authors entertained the hypothesis that KRAS is directly modified by ROS. They focused on H95 because they mapped the effect to the G domain and noted that H95 is a solvent-exposed, oxidation-susceptible residue in KRAS that does not occur in HRAS. PM association and nanoclustering of an H95A mutant was unaffected by FeCINO and a reciprocal Q95H substitution in HRAS rendered it sensitive. The authors conclude that FeCINO inhibits KRAS membrane association and signaling by oxidation of H95.

The study is well designed, well performed and clearly presented. The findings are of interest to RAS biologists and may have translational potential.

The major weakness is that the authors do not provide direct evidence of FeCINO-induced oxidation of H95. The mutational analysis is indirect and does not prove oxidation.

However, obtaining direct evidence of histidine oxidation by mass spectroscopy can be challenging (PMID: 19160434). If oxidized histidine can be detected, determination of the stoichiometry of modification will be important to explain how the modification affects signaling.

We thank R3 for the great comment and suggestion, which we fully agree with. Unfortunately, electron-transfer dissociation (ETD) to address oxidation of histidine at our university is unavailable, and collision-induced dissociation (CID) was the best option we had to address this important question. Our CID data indirectly suggest that His95 is oxidized by ferrocene and H₂O₂. This is shown in Fig. 5J and discussed in lines 338 – 348.

Minor points:

On page 4 the authors write that "PDE6 δ binds endocytosed Ras proteins via the farnesyl moiety." Endocytosis is not part of the process through which PDE6 δ extracts farnesylated proteins from membranes.

We thank R4 for point out the mistake. This now states, "PDE6 δ continuously sequesters Ras proteins from endomembranes via interacting with the prenyl moiety and returns them to the PM." in lines 97 – 99.

The effects of FeCINO on the growth of KRAS mutant and wild-type cells shown in Fig. 1 should be shown as a dose response, at least for a subset of sensitive and insensitive cell lines.

The suggested dose response curves for PDAC and NSCLC are shown in Fig. 1B and D with IC₅₀s for each cell lines. This is discussed in lines 155 – 160.

August 21, 2023

RE: Life Science Alliance Manuscript #LSA-2023-02094-TR

Dr. Kwang-Jin Cho
Boonshoft School of Medicine, Wright State University, Dayton
3640 Colonel Glenn Hwy
Dayton 45435

Dear Dr. Cho,

Thank you for submitting your revised manuscript entitled "A new ferrocene derivative blocks K-Ras localization and function by oxidative modification at His95". We would be happy to publish your paper in Life Science Alliance pending final revisions necessary to meet our formatting guidelines.

- please add ORCID ID for the corresponding author -- you should have received instructions on how to do so
- please upload all figure files as individual ones, including the supplementary figure files; all figure legends should only appear in the main manuscript file
- please add the Twitter handle of your host institute/organization as well as your own or/and one of the authors in our system
- please consult our manuscript preparation guidelines <https://www.life-science-alliance.org/manuscript-prep> and make sure your manuscript sections are in the correct order
- please add your main and supplementary figure legends to the main manuscript text after the references section;
- please include all authors in the Author Contribution section of your manuscript.
- please upload one figure per file
- LSA allows supplementary figures, but not EV Figures; please update your callouts for the Supplementary Figures in the manuscript Fig EV1A = Fig S1A)
- please add callouts for Figure S5A-D to your main manuscript text

Figure Check:

- the blots in Figure S4 appear to be not continuous. Please provide source data.

A. FINAL FILES:

B. MANUSCRIPT ORGANIZATION AND FORMATTING:

Sincerely,

Reviewer #1 (Comments to the Authors (Required)):

The authors have sufficiently addressed my comments.

August 24, 2023

RE: Life Science Alliance Manuscript #LSA-2023-02094-TRR

Dr. Kwang-Jin Cho
Boonshoft School of Medicine, Wright State University, Dayton
3640 Colonel Glenn Hwy
Dayton 45435

Dear Dr. Cho,

Thank you for submitting your Research Article entitled "A new ferrocene derivative blocks K-Ras localization and function by oxidative modification at His95". It is a pleasure to let you know that your manuscript is now accepted for publication in Life Science Alliance. Congratulations on this interesting work.

DISTRIBUTION OF MATERIALS:

Again, congratulations on a very nice paper. I hope you found the review process to be constructive and are pleased with how the manuscript was handled editorially. We look forward to future exciting submissions from your lab.

Sincerely,
